# The Influence of the Matrix Selection and the Unification Process on the Key Parameters of the Conductive Graphene Layers on a Flexible Substrate

**DOI:** 10.3390/ma16031238

**Published:** 2023-01-31

**Authors:** Sandra Lepak-Kuc, Łukasz Nowicki, Daniel Janczak, Małgorzata Jakubowska

**Affiliations:** 1Institute of Metrology and Biomedical Engineering, Faculty of Mechatronics, Warsaw University of Technology, Sw. Andrzeja Boboli 8, 02-525 Warsaw, Poland; 2Centre for Advanced Materials and Technologies (CEZAMAT), Warsaw University of Technology, 02-822 Warsaw, Poland

**Keywords:** graphene pastes, screen printing, triple roll process, carriage

## Abstract

Screen-printed graphene layers on flexible substrates are one of the most advanced printed electronics developments of recent years. Obtaining thin, flexible, highly conductive components, whose applications are increasingly directed towards biomedical engineering and even medicine, requires an in-depth understanding of the correct choice of materials and procedures. Our work was aimed at investigating the influence of homogenisation in the triple rolling process over pastes dedicated to the screen printing technology, on their rheological parameters and the properties of the prints. The effect of selecting a suitable polymer matrix and different packing of graphene flakes was evaluated. Several studies were carried out, which can provide an excellent knowledge base in the context of graphene screen-printing pastes. Paste rheology, printability, path thickness, sheet resistance and adhesion to the substrate were investigated. Selected layers were also subjected to SEM imaging.

## 1. Introduction

One of the fastest-growing industries in recent times is printed electronics. The possibility of obtaining components with high mechanical resistance while maintaining flexibility has led to an increase in research in this area [1]. Printed electronics can be used in many fields that we are already deal with every day. The variety of printed electronics solutions enables a wide range of applications. These include, for example, the mobile phone industry [2,3], smart clothing [4], and RFID tags [5]. Rapidly developing technology related to biomedical engineering or flexible substrates has resulted in the need for more precise machines and higher demands being placed on the materials produced [4].

The most versatile and extensively employed technology for printed electronics is screen printing. This technology is not only characterised by its high versatility as well as a good resolution but also enables printing on a large variety of substrates [6,7]. Screen printing has been successfully used to manufacture electrodes, capacitors, resistors, and sensors [8,9]. Depending on the printed material, the type of screen and the squeegee speed, print thicknesses of 10 to 100 μm are achievable [10]. Such a thickness allows a relatively good electrical conductivity to be reached compared to other printing techniques [11]. 

Screen printing technology imposes specific requirements on the materials involved. A paste form is required, i.e., a stable suspension of nano- to micro-sized particles in a polymer matrix. The properties of the pastes are adjusted to the dedicated application. However, it is not only the characteristics of the final printed product that are important but also the ease and repeatability of the printing process [5].

One of the breakthroughs, in terms of the expansion of prospective screen-printing technology applications, was the implementation of printable graphene-based composites. Based on these materials, these include innovative electrodes [12], capacitors [5] or biomedical sensors [13,14]. Through the research carried out, paths were achieved using screen printing, which had a resistivity of 1.04 · 10^−5^ Ω∙m [15]. A graphene-based RFID device was also developed. The paths that were obtained here had a sheet resistance of 1.4 Ω/□ [16]. 

Nevertheless, it should be highlighted here that for a value-enhanced application, it is not only the material of the functional phase that is crucial but also the carrier used. The right polymer can significantly improve the uniformity of the paste and provide the right properties for the composite to be used in the printing process. The polymer serves as a matrix in the final print. It is intended to provide the best possible mechanical performance, such as surface hardness, high abrasion and scratch resistance, as well as chemical resistance. Considering printing on a flexible substrate, it is also crucial to ensure adequate flexibility. Popular and widely used polymers as matrices are plastics (polycarbonates, thermoplastic polyesters or polyamides) or thermosetting and chemically curing resins [17,18]. Examples of such polymers are polylactide acid (PLA) and ABS (poly(acrylonitrile-co-butadiene-co-styrene)), often used as filaments in 3D printing [19], PMMA used to produce acrylic glass (commonly known as Plexiglass) [20]. or thermoplastic polyurethane (TPU) known from the manufacturing of phone protectors.

The increasing pressure to produce increasingly efficient layers, often both highly conductive and flexible, enabling ease of printing and proper adhesion to the substrate, often entails complex and multi-step paste manufacturing processes. One of the often-mentioned paste production steps is its homogenization in the rolling process [21,22]. This process can affect various aspects of both pastes and printed layers differently, but there have not been many studies dedicated to the impact of this process [20,23].

Furthermore, very often engineers and scientists are faced with the dilemma of the necessity to compromise between printability and mechanical and electrical properties. Therefore, investigating the impact of the triple-rolling process on various parameters can provide precious information for many researchers and manufacturers. In this paper, we present a study on the effects of the rolling process of graphene pastes based on different carriers on their rheology, printability and electrical and adhesion parameters. Flexible substrates, which are among the most researched at the moment due to the rapid growth of flexible electronics, were investigated in this study. The investigation was dedicated to graphene pastes, which represent a strong trend in the development of screen-printing technology.

## 2. Materials and Methods

### 2.1. Materials

#### 2.1.1. Graphene Flakes

In this work, graphene nanoplatelets from CheapTube^®^ (Cambridgeport, VT, USA) were used. According to the datasheet, they range from 1 to 2 μm in width and 8–15 nm in thickness.

#### 2.1.2. Carriage

In this study, four different polymers were used. The solvents were selected concerning the particular polymers tested, ensuring good solubility and with consideration of their use in screen-printing technology, i.e., paying attention to an appropriately high boiling point. The first polymer used was poly(methyl methacrylate) with an average molecular weight of 3.5 · 10^5^ u, which dissolved in 2-(2-butoxyethoxy) ethyl acetate to form a carrier referred to in this paper as PMMA. Another polymer tested was Laroflex M35 with a density of 1.24 · 10^3^ kg/m^3^, which was dissolved in a mixture of 2-(2-butoxyethoxy) ethyl acetate and 2-butoxyethanol in a ratio of (97:3); referred to as LARO carrier. Two types of thermoplastic polyurethane (TPU) were also tested, in the form of Elastollan^®^ soft 35AP, with a density of 1.18 · 10^3^ kg/m^3^, and Elastollan^®^ hard C80A, with a density of 1.19 · 10^3^ kg/m^3^. Both were dissolved in N, N-Dimethylformamide; their designations in the paper are TPU soft and TPU hard.

Poly(methyl methacrylate) and all solvents were purchased from Sigma Aldrich^®^ (Darmstadt, Germany), Laroflex M35, Elastollan soft 35AP and Elastollan C 80A were purchased from BASF^®^ (Ludwigshafen, Germany).

The printing substrate was a 36 µm PET film purchased from MICEL Sp z o.o (Zychlin, Poland).

Silver contacts were applied with LOCTITE^®^ ECI 1010 ink purchased from Tekra LLC (New Berlin, WI, USA).

### 2.2. Methods

#### 2.2.1. Pastes Preparation

The carriers were obtained by mixing the weighed components for 48 h at a temperature adapted to the individual polymers comprising the carriers, 50 °C for LARO and PMMA and 30 °C for soft and hard TPU, respectively.

In the next step, the carriers were combined with nanoplatelets of graphene, representing the conductive functional phase. An appropriate amount of graphene and carrier was weighed and ground for at least five minutes in an agate mortar. Each series was prepared in five different degrees of packing of the functional phase (5%, 7.5%, 10%, 12.5% and 15%, respectively. We have prepared pastes with 5–15% packing since pastes with this amount of graphene possess an adequate viscosity for screen-printing technology. Each of the resulting pastes underwent a rolling process using an EXAKT model 80E laboratory three-roll mill (EXAKT Advanced Technologies, Norderstedt, Germany) (Figure 1) with a set gap of 5 µm between the rollers and a torque of 0.2 N/mm.

#### 2.2.2. Printing Process

The paths were printed using an Aurel c920 screen printer (Figure 2), Aurel Automation (AUREL s.p.a., Modigliana, Italy) on a flexible PET film containing pre-applied silver contacts. Prints were dried in an SLW 115 STD dryer from POL-EKO at 115 °C for 20 min. The drying parameters have been adjusted so that the prints are well dried, that is, to evaporate the solvent without liquefying the polymer matrix, i.e., melting the tracks on the substrate. A polyester screen with 77T mesh was used. 

#### 2.2.3. Rheology Testing

The rheology of the pastes was measured with a Brookfield^®^ R/S-CPS+ rheometer equipped with an RCT-50-2 spindle (AMETEK Brookfield, DE, USA), dedicated to the viscosity ranging between 0.006–50,900 Pa·s. A shear rate from 0 to 400 s^−1^ for was tested for 100 s. The viscosities were analysed using the dedicated Rheo 3000 software (version 1.2.2009.1). 

#### 2.2.4. Electrical Testing

The resistance of the printed paths was measured with a UT804 multimeter manufactured by Uni-Trend Technology^®^ (Shanghai, China). The values were recalculated into sheet resistance (Ohms per square).

#### 2.2.5. Paths Thickness 

Path thickness was measured using Bruker’s DektakXT^®^ profilometer (Bruker, Billerica, MA, USA), with the force of the stylus set to 3 mg.

#### 2.2.6. Adhesion Testing

Adhesion was tested using a 2 mm ANTICORR^®^ disc knife included in the BG VF 1842 C kit. Scotch^®^ tape (ANTICORR^®^ Gdansk, Poland) was applied to the cut sections and ripped off after approximately three minutes by uniform movement. The crumbled material was removed with a brush and the results were compared according to the ISO 2409 standard, which defines the level of adhesion on a scale of 0–5, where 0 means no material detachment at all, 1 indicates from 0–5% of the total material, 2 from 5–15%, while 3 is between 15–35% and 4 is between 35% and 65% of the total pattern. Larger material detachment is qualified as 5. 

#### 2.2.7. Microscopy

Scanning electron microscopy (SEM) of three layers of best conductivity was conducted on a Hitachi SU8230 instrument (Tokyo, Japan) with an accelerating voltage of 5.0 kV and an upper secondary electron detector.

For the obtained pathways with markedly different conductivity capabilities, their structure was additionally compared under the Keyence VHX-900F optical microscope (Keyence Corporation, Osaka, Japan).

## 3. Results and Discussion

### 3.1. Viscosity Tests

Firstly, the dependence of the pastes’ viscosity on the shear rate was analysed. This comparison was carried out for all the tested carriers and graphene concentrations, in all cases for both triple-rolled and non-rolled materials. Figure 3 shows exemplary results, selected to illustrate the dependence between viscosity and shear rate, due to the change in graphene packing for unrolled pastes (Figure 3a), rolled pastes (Figure 3b), the viscosity change caused by the rolling process for a chosen carrier (Figure 3c) and concentration, and for a chosen concentration of the functional phase, comparing all tested carriers (Figure 3d). The rheology of the pastes depends on the dimensions, the uniformity of the dimension distribution, and the purity of the specific graphene flakes. Thus, the dependence of the presented results can be translated to other graphene materials, but not the specific values obtained.

For all four carriers tested, higher dynamic viscosity was observed with increasing graphene packing for both triple-rolled and non-rolled materials. A tripling of the graphene concentration increased the dynamic viscosity registered for the minimal shear rate by up to several hundred times. This is understandable, as graphene nanoplatelets are a powder additive with a highly developed specific surface area and low bulk density. Furthermore, for all pastes, irrespective of the carrier used, as the packing increased, a greater curve slope angle was recorded for shear rates below 150 1/s. For higher speeds, the correlation between viscosity and the aforementioned parameter decreased dramatically, i.e., a further increase in shear rate resulted in a slight change in viscosity.

Considering the effect of the rolling process on the rheological properties, various dependencies can be observed. For PMMA pastes, an improvement in rheological properties and a decrease in dynamic viscosity were noted for all pastes. For LARO pastes, the rolling process resulted in a decrease in viscosity for the packing of graphene by 15% and an increase for the other variants. For pastes containing soft TPU, the rolling process caused a decrease in the viscosity for pastes with higher packing (12.5% and 15%). Such a result indicates that for those carriers only for larger packing, the rolling process introduces a significant deagglomeration of the functional phase and contributes to a reduction of internal friction in the material which results in lower viscosity. The hard TPU-based pastes showed worse rheological properties after the rolling process. For shear rates above 100 1/s, the dynamic viscosity values decreased rapidly and were close to zero. 

Comparing the results obtained for various carriers, the highest dynamic viscosity was observed for the hard TPU paste (Figure 1d)). Analysing the viscosity dependence on shear rate, one can see that apart from hard TPU, the course for the other three carriers is comparable and the observed values for rates above 100 1/s were not significantly different. 

In screen printing technology, the standard dynamic viscosity is between 0.5–50 Pa∙s for the shear rate achieved during squeegee work (This speed is 150–180 1/s). Therefore, despite significant differences and sometimes extremely high viscosities at the zero shear rates, almost all of the pastes examined showed adequate viscosity for the speeds relevant for use in screen printing. The exceptions are rolled pastes based on hard TPU, where viscosity was too low.

### 3.2. Resistance Tests

Next, the electrical properties of the tested materials were compared. For this purpose, the sheet resistance of a dozen rectangular paths of different thicknesses printed on dedicated silver contacts was measured. During the process of preparing the paths, significant difficulty in the printing process was noted for pastes containing hard TPU. Their high viscosity, combined with strong adhesion to the squeegee and screen, inhibited proper deposition of the paste on the substrate, resulting in frayed and patchy paths (Figure 4). 

Such a result qualifies the TPU hard carrier as not screen-printable. For this reason, further research on this material was discontinued. In addition, non-rolled pastes containing soft TPU showed limited printability. At higher graphene packing (12.5% and 15%), the material failed to settle well on the substrate despite several passes of the squeegee. Since only part of the material was squeezed through the screen in a single printing cycle, which would not be optimal for the screen-printing process, it was decided to set the maximum graphene packing of this carrier at 10% by weight without a triple-rolling process. Adding the rolling step enabled printing and the higher tested graphene packing. 

The results obtained for the pastes, found to be highly screen-printable, are presented in Table 1. For each paste, resistance measurements were carried out 24 times and then the average was calculated.

For both rolled and unrolled pastes, with all three carriers considered, a decrease in the layer resistance of the paths was observed as the graphene packing increased. This is a predictable result because graphene is the conductive functional phase of these materials. For LARO-based pastes, with a threefold increase in graphene packing, the sheet resistance decreased by up to two orders of magnitude. 

When comparing the paths printed from pastes subjected to the unification process on the triple-roller with those not subjected to this process, it is worth noting that in most cases a benefit in terms of a reduction in sheet resistance is evident. The greatest changes were observed for the low graphene packing. For LARO pastes with the lowest graphene content, rolling resulted in a more than 100-fold decrease in sheet resistance. For LARO pastes containing the highest tested packing, there was a decrease in the layer resistance value of 0.12 kΩ/□. 

For the pastes on soft TPU, rolling not only improved the bonding of the functional phase to the polymer phase and unified the material enough to make it printable, but an improvement in electrical properties is also apparent. 

For PMMA pastes, the rolling process only had a positive effect on pastes with up to 10% packing. In contrast, the 12.5% and 15% pastes had a slightly higher electrical resistance after the rolling process.

The reason for the diminishing effect of improving electrical parameters with increasing packing can be attributed to the fact that, with excessive amounts of graphene, the rolling process can lead not only to deagglomeration but also to fragmentation of the graphene flakes, which in turn can result in more difficult carrier tunnelling and thus poorer electrical properties.

Comparing all of the carriers, it is interesting to observe that, at low graphene packing, significant differences in the obtained values of sheet resistance can be noted, while for maximum graphene packing the resistance results are very similar for all the carriers. The interactions between the filler particles and the matrix polymer seem to be more crucial for low functional phase contents. For these packings, an easy path for electron tunnelling between flakes is important, so polymer density and particle size may also be significant. At higher packing levels, the role of the polymer is more directed towards maintaining the print on the substrate and enabling the material to maintain homogeneity.

### 3.3. Path Thickness Tests

An important parameter that is very often neglected in investigations, but which may vary depending on the packing of the functional phase, and also on the carrier used, is the path thickness. This parameter has great significance in the context of the obtained resistance of the layers. We examined the path thicknesses of all the paths obtained (Table 2). The thickness was measured for four paths of different widths and an average was calculated.

The results obtained show that, as the packing increases, the printed paths have a higher thickness. That may be directly related to the viscosity of the pastes, as pastes with lower viscosity spread over the substrate to a greater extent. It is also noticeable that LARO pastes allow for thicker paths to be printed, which may be attributable to the amount of polymer required to produce a paste with suitable rheology, but also to the higher intermolecular interactions within this carrier. However, the influence of the carrier itself is considerably lower than the packing of graphene flakes, which is understandable, given that the amount of the added functional phase is the main determinant of paste viscosity.

Non-rolled pastes have been omitted from this comparison due to their very similar outcomes, indicating that the path thickness is not significantly affected by the triple roll process.

### 3.4. Adhesion Tests

In the final stage of the study, the adhesion properties of the printed paths to the flexible PET substrate used were checked for both rolled (R) and not rolled (NR) pastes. In this work, we used flexible substrates on account of the strong development of flexible electronics. It is on difficult, flexible substrates that additional material processing, in the form of three-roll milling, can allow graphene pastes to be printed with adequate adhesion. Tests were carried out for at least three 20 mm × 20 mm square prints each time and the median of the adhesion values obtained is listed (Table 3).

The definite effect on adhesion to the substrate for both the polymer matrix used and the concentration of the functional phase is evident. The more graphene in the layer, the better the adhesion for all three polymer matrices tested.

The poorest adhesion properties among the tested polymers were noted for PMMA-based prints. For the 5% graphene packing, 35–65% detachment was observed. However, for higher concentrations of graphene, the detached material did not exceed a value of 5% of the print. 

Prints on both PMMA and TPU matrixes showed similar adhesion. In the most severe cases, this resulted in the detachment of no more than 15% of the print (value 2 on the adhesion scale in the standard used)

Such dependencies can originate from several factors; from the properties of the polymers themselves, such as their elasticity and density, the interaction of the used solvent with the PET substrate and the interaction of the carrier with the functional phase, finishing with the viscosity of the paste modulated by both the carrier and the graphene packing.

In addition, the effect of the rolling process of the pastes on the adhesion of the layers printed with their use is also evident. This is probably related to the homogenisation of the paste itself. Since the agglomerates of the functional phase, which are present in greater quantity in non-rolled pastes, can be the link that detaches the layer from the substrate, the deagglomeration, being part of homogenisation, leads to better adhesion.

### 3.5. Scanning Electron Microscopy

With the highest packing of graphene being 15%, for pastes based on all three carriers subjected to the unification process on a three-roller machine, the results of layer resistance, layer thickness and adhesion were found to be very similar. These layers were subjected to SEM imaging to verify to what extent the structures of these prints are alike (Figure 5).

The analysis of the SEM images shows that, despite the differences in the polymer matrix, with sufficiently high packing of the functional phase, highly similar layers are obtained. The homogeneous and dense distribution of graphene flakes is evident. The visible bulges are the result of the screen-printing technology used.

## 4. Conclusions

The role of both the composition and the selection of the process for producing graphene-based screen-printing pastes was outlined. The importance of the selection of the polymer and the degree of packing of the functional phase was discussed, as well as the influence of the triple-rolling process concerning rheological parameters, screen-printing potential, and properties of the obtained paths concerning sheet resistance, path thickness, and adhesion to the flexible substrate.

Depending on the carriers used, different rheological, electrical and adhesion properties of the composites produced were obtained. It was shown that the results can be significantly different even within a single polymer family, as in the case of the two thermoplastic polyurethanes tested, both of which were based on polyethers, and differing in hardness and the presence of a plasticiser (in soft TPU).

We have shown that graphene pastes with a higher packing of the functional phase, possessing higher dynamic viscosity values, not only exhibit better electrical performance but also have better adhesion properties to PET films.

Above all, we demonstrated the significance of a correct paste preparation process. The triple-rolling process revealed the influence on the rheology and printability of the pastes, as well as the sheet resistance and adhesion of the acquired paths. The rolled materials showed lower electrical resistivity, with the greatest impact at lower graphene packing. A clear improvement in the adhesion of the rolled composites was noted for all carriers.

## Figures and Tables

**Figure 1 materials-16-01238-f001:**
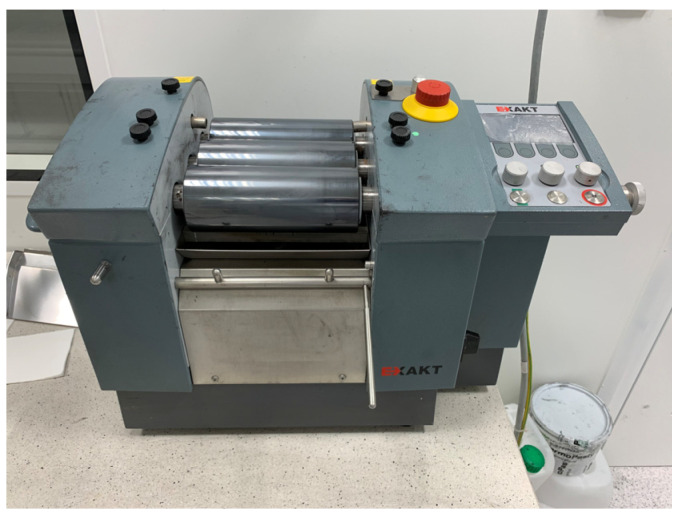
Three-roll mill EXAKT model 80E.

**Figure 2 materials-16-01238-f002:**
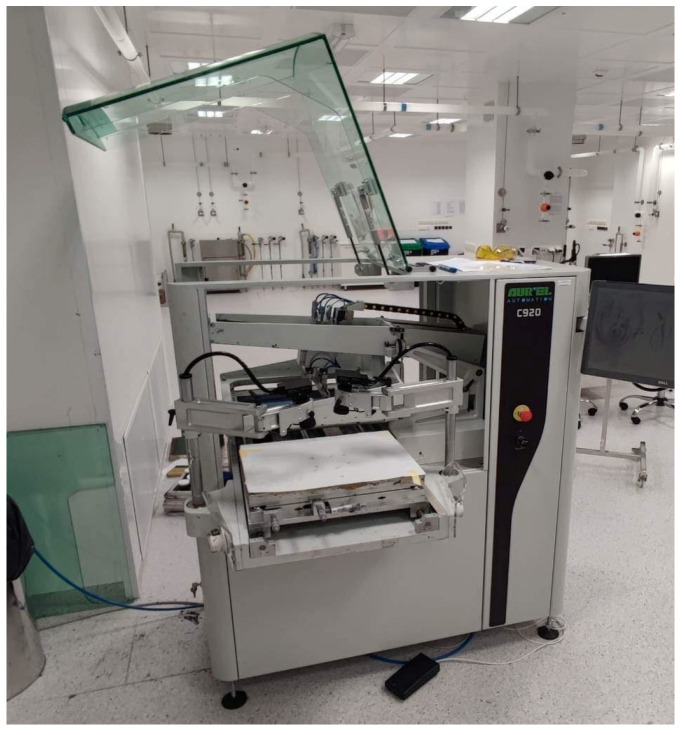
An Aurel c920 screen printer.

**Figure 3 materials-16-01238-f003:**
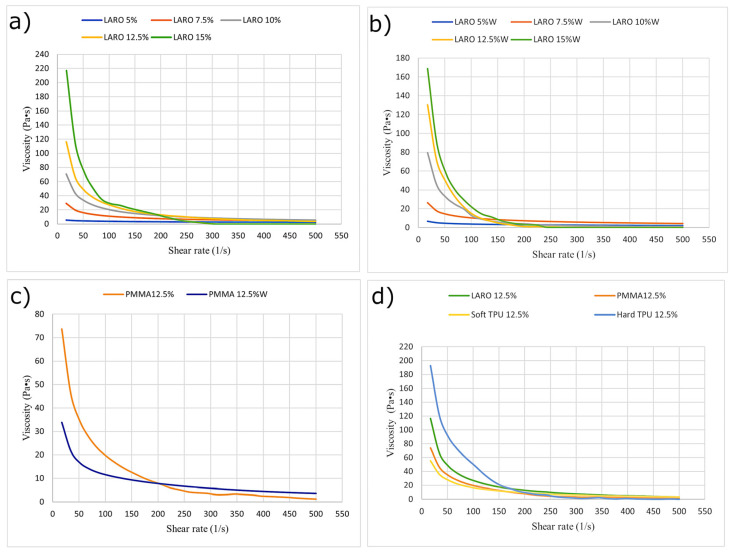
Reliance between viscosity and shear rate for non-rolled LARO pastes (**a**); rolled LARO pastes (**b**); 12.5% PMMA paste before and after the triple rolling process (**c**); pastes of various carriers containing the same graphene amount (12.5%) (**d**).

**Figure 4 materials-16-01238-f004:**
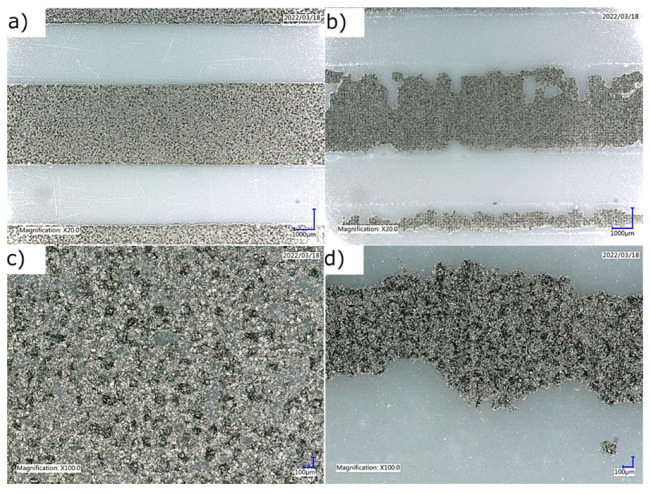
Comparison of microscopy images for two magnifications 20× and 100× of prints obtained for LARO paste (**a**,**c**) and hard TPU paste (**b**,**d**).

**Figure 5 materials-16-01238-f005:**
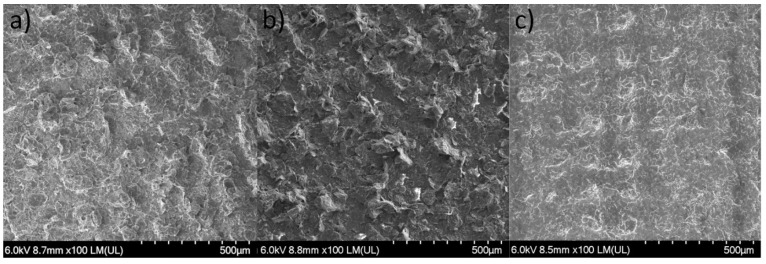
Comparison of SEM images of layers containing 15% packing of graphene based on LARO (**a**), PMMA (**b**), and soft TPU (**c**).

**Table 1 materials-16-01238-t001:** The sheet resistance of paths printed with tested graphene pastes. NR means non-rolled, R-rolled.

Graphene Packing	Sheet Resistance (kΩ/□)					
	LARO		PMMA		Soft TPU	
	NR	R	NR	R	NR	R
5%	46.8	9.74	30.52	5.75	18.90	5.64
7.5%	3.95	2.11	3.59	3.05	2.70	1.64
10%	1.71	0.58	0.87	0.30	0.81	0.51
12.5%	0.37	0.33	0.37	0.38	-	0.36
15%	0.33	0.21	0.25	0.26	-	0.26

**Table 2 materials-16-01238-t002:** Path thickness of all printed paths.

Graphene Packing	Average Path Thickness (µm)		
	LARO	PMMA	Soft TPU
5%	7.68	3.89	5.44
7.5%	11.47	7.58	8.13
10%	12.43	14.72	13.80
12.5%	16.89	15.45	17.43
15%	21.31	19.86	19.03

**Table 3 materials-16-01238-t003:** Adhesion of the printed paths to the flexible PET film.

Graphene Packing	Adhesion					
	LARO		PMMA		Soft TPU	
	NR	R	NR	R	NR	R
5%	2	1	4	4	2	1
7.5%	2	0	3	3	1	1
10%	1	0	2	1	0	0
12.5%	1	0	2	1	-	0
15%	0	0	1	1	-	0

## Data Availability

Not applicable.

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
