# Peer review of "The Influence of the Matrix Selection and the Unification Process on the Key Parameters of the Conductive Graphene Layers on a Flexible Substrate"

_materials, 2023, doi:10.3390/ma16031238_

Round 1
Reviewer 1 Report
I recommend adding a more detailed description of the quality analysis method, complete with pictures or diagrams describing the measurement principle.
Author Response
At the very beginning, we would like to thank the Reviewer for his valuable suggestion allowing us to improve the manuscript.
Comment 1: I recommend adding a more detailed description of the quality analysis method, complete with pictures or diagrams describing the measurement principle.
Thank you very much for this suggestion. We added photos of the three-roll mill and screen-printing machine.
Reviewer 2 Report
Overall, well documented work. This is a well-organized, multi-parameter study.
I have only few minor comments/suggestions.
In printing process section, 0C not typeset correctly in line 117.
The word frictions should be friction line 177
Tables 2 and 3 should not be split on different pages.
In Table 3 one should add meaning of R and NR as in Table 1. Also, it would be better to give the meaning of integers in table description for results.
What is lacking in the present study was any kind of statistics on repeatability of properties which is also of great importance in printed electronics. (Only thickness measured four times yet electrical resistivity and adhesion has no mention of repeated measurements as I could tell.) Given the relatively fast ability to do multiple prints, I'm a bit surprised this was not done. If it was done, please comment on that in methods or results section. Still, I feel the conclusions are still relevant and has impact on future studies of printed electronics.
Author Response
At the very beginning, we would like to thank the Reviewer for the insightful and valuable comments and suggestions for the manuscript.
Comment 1:
- In printing process section, 0C not typeset correctly in line 117.
Thank you for pointing this out, the mistake is fixed.
Comment 2:
- The word frictions should be friction line 177
Thank you for pointing this out, this mistake is fixed.
Comment 3:
- Tables 2 and 3 should not be split on different pages.
We are very grateful for this suggestion. We have improved the structure of the article and made sure that all tables are not split on different pages.
Comment 4:
- In Table 3 one should add meaning of R and NR as in Table 1. Also, it would be better to give the meaning of integers in table description for results.
We are very grateful for this suggestion. We would like to inform that we added the explanation in the subsection 3.4
Comment 5:
- What is lacking in the present study was any kind of statistics on repeatability of properties which is also of great importance in printed electronics. (Only thickness measured four times yet electrical resistivity and adhesion has no mention of repeated measurements as I could tell.) Given the relatively fast ability to do multiple prints, I'm a bit surprised this was not done. If it was done, please comment on that in methods or results section.
Thank you very much for raising such an important issue. We added information about repeated resistivity measurements in line 227. The information about the number of measurement repetitions of adhesion is given in line 291.
Author Response
At the very beginning, we would like to thank the Reviewer for the insightful and valuable comments and suggestions for the manuscript.
Comment 1:
- Could the proposed analysis of rheology, solid loading, type of polymers be generalised and used for graphene nanoplatelets-based inks in general, or is it very dependent and related to the specific commercial product used in this study?
Thank you, for your insightful question. The rheology of the pastes, which entails printability and subsequent measurement considerations, depends on the dimensions, the uniformity of the dimension distribution, and the purity of the specific graphene flakes.
According to our long-term research in the field of graphene-based screen-printed pastes, the commercial material used has high uniformity and good electrical properties.
Comment 2:
- What is the main motivation and necessity behind using the polymer carrier in this publication? The literature reports on various pastes based on high pristine graphene loading concentrations, by which the screen-printed patterns are achieved with much higher electrical conductivity (by using only solvent, dispersant and rheological agent).
Thank you for this comment. We added clarification in the article on the role of polymers as carriers in lines 56-60: "The polymer serves as a matrix in the final print. It is intended to provide the best possible mechanical performance, such as surface hardness, high abrasion and scratch resistance, as well as chemical resistance. Considering printing on a flexible substrate, it is also crucial to ensure adequate flexibility."
Comment 3:
- How was the packing of graphene selected to be in the range 5%-15%? Multiple Tables (e.g. 1 and 2) could be better and much clearly presented in a graphical form, as the dependences on the graphene contents. What is the accuracy (and the deviations) of the sheet resistance and thickness measurements (measured as precise as with 2 decimal places)?
Thank you for this suggestion. We decided to use tables instead of graphical form to present all our results together. Trying to compare the results depending on the type of carrier, rolling process and packing would require using several independent graphics because putting it on one graphic could make it look difficult to read. The choice of packing was made based on our experience in working with pastes for screen printing applications. Pastes with higher packing than 15% have too high viscosity and are unsuitable for screen printing. Pastes with lower packing had too low viscosity and spilt in an undesirable way over the substrate. We updated the manuscript with this information.
Comment 4:
- Figure 2 specifically should contain the photo/micrographs of all the obtained and tested sample types (for LARO, PMMA, Soft TPU, Hard TPU). Figure 3 does not demonstrate any clear observation of the findings with the currently used magnification. Not only the surface structure should be considered and investigated, but also within the layer thickness (cross-sections).
Thank you for this suggestion. The main purpose of putting this figure in the manuscript was to underline the reliance between adhesion, viscosity, and sheet resistance by showing the structure of printed paths. We could add photos to compare all pastes, but it won’t provide any new data than we obtained from conducted measurements.
Comment 5:
- It is known that the duration of annealing process and annealing temperature of the printed paths (mentioned in the paper as drying process with fixed parameters) are also playing an important role in the final performance of conductive patterns. The selection of the drying/annealing step parameters should be investigated or at least commented.
Thank you for this comment. We agree that the drying process affects the properties of the layers. The parameters used in work are based on our experience of working with similar materials. We added an explanation in lines 128-130.
Comment 6:
- The motivation to use flexible substrate in this study is unclear and along with the adhesion tests of the printed paths to the flexible PET film, also flexing/stress tests should be considered, otherwise, there is no clear evidence on the mechanical stability of the prints, and any other rigid substrate can be used.
Thank you for bringing this issue to our attention. In this article, we wanted to focus primarily on the impact of the rolling process and how important the choice of the polymer matrix is in the context of the properties of the screen printing pastes and the paths printed from them. We used flexible substrates on account of the strong development of flexible electronics. It is on difficult, flexible substrates that additional material processing, in form of three-roll milling, can allow graphene pastes to be printed with adequate adhesion. While we agree with the statement that if we wanted to test how the material behaves under bending or stretching, a bend/stretch test should be carried out, it is the issue of adhesion that is crucial during such tests, and this was tested in the paper.
Comment 7:
- While the clarity of the text is fair, English language editing is suggested to make the text more readable and ensure it is clear in terms of the main idea and overall impact of the study.
Thank you for this suggestion. We would like to inform you that manuscript went through extensive language revision.
Comment 8:
The Conclusions section seems to be too general and vague, there is no clear understanding of main findings and impact of the study from it. The same comment is applied to the Abstract of the manuscript.
Thank you for these suggestions. We have made refinements within both the abstract and the conclusions.
Reviewer 4 Report
This manuscript presented a study on the effect of the level of graphene pastes in composites. There are some issues for possible publication in this journal.
1. The novelty and advance of this manuscript are not enough, the study is still limited, and more data on the experiment should be given.
2. The paper is not well organized, missing some explanation. For example, in Table 1, what are LARO, NR, R, PMMA? Please write their full names for the first time the author mentions them.
3. The author should work on the introduction and explain why this work provides new knowledge or innovation to the field. All experiments in the paper are quite simple.
4. Please pay attention to the consistency of the tense, grammar, and other minor formatting problems throughout the written manuscript.
Above all, I think this manuscript does not match the characteristics of Materials journal.
Author Response
At the very beginning, we would like to thank the Reviewer for the insightful and valuable comments and suggestions for the manuscript.
Comment 1:
- The novelty and advance of this manuscript are not enough, the study is still limited, and more data on the experiment should be given.
We are grateful for the concern about the conclusions and number of experiments presented in our work. We are certain that the number of experiments conducted is comparable to or even higher than in other publications in this field.
- Cruz, S.; Rocha, L.; Viana, J. Printing Technologies on Flexible Substrates for Printed Electronics. In; 2018 ISBN 978-1-78923-456-5.
- Kumar, S.; Kumar, P.; Bhatt, K.; Shrivastva, S.; Kumar, A.; Singh, R.; Punia, R.; Tripathi, C.C. Impact of Triple Roll Milling Processing Parameters on Fluidic/Rheological and Electrical Properties of Aqueous Graphene Ink. Advanced Engineering Materials 2020,
Comment 2:
- The paper is not well organized, missing some explanation. For example, in Table 1, what are LARO, NR, R, PMMA? Please write their full names for the first time the author mentions them.
We are very grateful for this suggestion. We would like to inform you that we added the explanation in line 232 (for R and NR). Explanations for LARO and PMMA are included in subsection 1.1 Materials. (Line 94 for PMMA and 96 for LARO).
Comment 3:
- The author should work on the introduction and explain why this work provides new knowledge or innovation to the field. All experiments in the paper are quite simple.
We really appreciate this comment. We would like to underline that we explain the importance of this research in conclusion. Our study includes knowledge which is important for other researchers to take into the consideration. It not only emphasizes how the choice of polymer alone affects the obtained parameters, but also how the properties of materials as well as printed paths can be significantly improved using the triple-rolling process.
Comment 4:
- Please pay attention to the consistency of the tense, grammar, and other minor formatting problems throughout the written manuscript.
Thank you for this suggestion. We would like to inform you that manuscript went through extensive language revision.
Round 2
Reviewer 3 Report
Some changes are implemented in the text, however, some of them are only explained in the response attachment, but not in the text.
The corrections are implemented, however, the overall quality of the presentation is still on the average level.
Author Response
Dear reviewer, we would like to express our appreciation for your efforts towards improving our manuscript. We have added the missing explanations in lines 169-172, and 294-296.
We are making a great effort to improve our work. A native speaker has linguistically checked the manuscript.
Reviewer 4 Report
I agree with the revision of the manuscript, so I recommend it for publication.
Author Response
Dear reviewer, thank you very much for your favourable opinion.